# Response of Nearby Sensors to Variable Doses of Nitrogen Fertilization in Winter Fodder Crops Under Mediterranean Climate

**DOI:** 10.3390/s25185811

**Published:** 2025-09-17

**Authors:** Luís Silva, Caroline Brunelli, Raphael Moreira, Sofia Barbosa, Manuela Fernandes, Andreia Miguel, Benvindo Maçãs, Constantino Valero, Manuel Patanita, Fernando Cebola Lidon, Luís Alcino Conceição

**Affiliations:** 1Earth Sciences Department, NOVA School of Science & Technology, Campus of Caparica, NOVA University Lisbon, 2829-516 Caparica, Portugal; svtb@fct.unl.pt (S.B.); fjl@fct.unl.pt (F.C.L.); 2VALORIZA—Research Center for Endogenous Resource Valorization, Polytechnic Institute of Portalegre, 7300-110 Portalegre, Portugal; luis_conceicao@ipportalegre.pt; 3Elvas School of BioSciences, Polytechnic Institute of Portalegre, Train Headquarters Building 14th of January Avenue n°21, 7350-092 Elvas, Portugal; 26138@ipportalegre.pt; 4Federal Institute of Education, Science and Technology of Espírito Santo, Campus Itapina, Espiríto Santo, Colatina 29717-000, Brazil; raphael.moreira@ifes.edu.br; 5GeoBioTec Research Center, NOVA School of Science & Technology, Campus of Caparica, NOVA University Lisbon, 2829-516 Caparica, Portugal; benvindo.macas@iniav.pt (B.M.); mpatanita@ipbeja.pt (M.P.); 6ADP–Fertilizantes, Apartado 88, 2616-907 Alverca do Ribatejo, Portugal; manuelafernandes@adp-fertilizantes.pt (M.F.); andreiamiguel@adp-fertilizantes.pt (A.M.); 7INIAV—National Institute of Agricultural and Veterinary Research, I.P., 7351-901 Elvas, Portugal; 8LPF-TAGRALIA, School of Agricultural, Food and Biosystems Engineering (ETSIAAB), Universidad Politécnica de Madrid, Avenida Puerta de Hierro 2-4, 28040 Madrid, Spain; constantino.valero@upm.es; 9Polytechnic Institute of Beja, Beja School of Agriculture, Rua Pedro Soares, 7800-295 Beja, Portugal; 10MED—Mediterranean Institute for Agriculture, Environment and Development and CHANGE—Global Change and Sustainability Institute, University of Évora, Mitra, Ap. 94, 7006-554 Évora, Portugal; 11InovTechAgro—National Skills Center for Technological Innovation in the Agroforestry Sector, 7300-110 Portalegre, Portugal

**Keywords:** real-time fertilization, automation, data analysis, monitoring, remote sensing, geographic information systems, sustainability

## Abstract

The sustainable intensification of forage production in Mediterranean climates requires technological solutions that optimize the use of agricultural inputs. This study aimed to evaluate the performance of proximal optical sensors in recommending and monitoring variable rate nitrogen fertilization in winter forage crops cultivated under Mediterranean conditions. A handheld multispectral active sensor (HMA), a multispectral camera on an unmanned aircraft vehicle (UAV), and one passive on-the-go sensor (OTG) were used to generate real-time nitrogen (N) application prescriptions. The sensors were assessed for their correlation with agronomic parameters such as plant fresh matter (PFM), plant dry matter (PDM), plant N content (PNC), crude protein (CP) in%, crude protein yield (CPyield) per unit of area, and N uptake (NUp). The real-time N fertilization stood out by promoting a 15.23% reduction in the total N fertilizer applied compared to a usual farmer-fixed dose of 150 kg ha^−1^, saving 22.90 kg ha^−1^ without compromising crop productivity. Additionally, NDVI_OTG showed moderate simple linear correlation with PFM (R^2^ = 0.52), confirming its effectiveness in prescription based on vegetative vigor. UAV_II (NDVI after fertilization) showed even stronger correlations with CP (R^2^ = 0.58), CPyield (R^2^ = 0.53), and NUp (R^2^ = 0.53), highlighting its sensitivity to physiological responses induced by N fertilization. Although the HMA sensor operates via point readings, it also proved effective, with significant correlations to NUp (R^2^ = 0.55) and CPyield (R^2^ = 0.53). It is concluded that integrating sensors enables both precise input prescription and efficient monitoring of plant physiological responses, fostering cost-effectiveness, sustainability, and improved agronomic efficiency.

## 1. Introduction

The response of nearby sensors to variable doses of nitrogen (N) fertilizer in winter fodder crops under a Mediterranean climate can be understood through the use of various sensor technologies and their impact on crop growth, N use efficiency (NUE), and environmental outcomes. These sensors measure crop N status; chlorophyll content; and, usually, the normalized difference vegetation index (NDVI) values. Studies have shown that N rate significantly impacts leaf N content and chlorophyll index, with NDVI values increasing throughout the season [1]. In winter wheat, sensors like the GreenSeeker^®^ have been used to measure NDVI at different growth stages, showing that sensor-based applications can be more efficient than traditional methods [2].

The Push-broom Hyperspectral Image Sensor captures reflectance in visible and near-infrared wavebands, which correlates with soil N and variable rate fertilization. It has been effective in detecting differences in soil N and managing fertilization to reduce growth variability in winter wheat [3,4].

Systems based on ISO 11783-compatible industrial sensors and control systems measure spectral responses to adjust N application rates in real time. These systems have shown positive effects on NUE and reduction of environmental risks by optimizing N distribution based on crop needs [5,6].

Variable rate application (VRA) of N has been shown to increase NUp and NUE by 58% compared to uniform application, although yield increases are not always significant [7]. In some cases, VRA has led to a reduction in N application by up to 38 kg ha^−1^ per year, improving NUE by 15% and reducing variability in N balances [8]. The use of sensors for VRA can reduce environmental pollution by minimizing N losses and optimizing fertilizer use. For instance, Medel-Jiménez et al. [9] showed an 8.6% reduction in CO_2_ emissions with VRA compared to conventional methods.

However, the effectiveness of sensor-based VRA can be influenced by spatial variability in soil properties and temporal factors such as weather conditions and crop development stages [10]. Accurate sensor calibration and understanding of site-specific conditions are crucial for optimizing N application and achieving consistent results across different fields and seasons [11]. While sensor-based VRA can improve NUE and reduce environmental impact, the high costs of sensors and their operation can be a barrier. Sharing or leasing sensor technology may be a viable solution for smaller farms [9].

The success of sensor-based variable rate N application in winter fodder crops under Mediterranean climates shows promise in improving NUE and reducing environmental impact; however, this depends on accurate calibration, understanding of site-specific conditions, and economic considerations [12,13,14]. This study aims to address these critical aspects by first validating cutting-edge on-the-go technology, which offers real-time artificial intelligence (AI) and VRA capabilities, setting it apart in precision agriculture. Secondly, we adopt an integrated and complementary approach, exploring the synergy between different sensors to enhance our understanding of crop response. Our third objective is to present efficiency results related to NUE, highlighting the impact on both sustainability and productivity. Finally, this work is conducted within the specific context of the Alentejo region and its winter fodder crops, providing a unique “laboratory” to validate the robustness of this technology under particular Mediterranean conditions.

## 2. Materials and Methods

The study’s methodology was structured into four sequential stages, as illustrated in the flowchart below (Figure 1). Stage 1 (experiment installation) focused on crop establishment, seeding annual ryegrass (*Lolium multiflorum* L.) under a no-till system, supplemented by basal fertilization and weed control. In Stage 2 (VRA N fertilization), variable rate N fertilization was carried out after on-the-go sensor installation and calibration. Fertilization rates varied between 50% of the normal dose (150 kg ha^−1^) and 100% of the normal dose. This stage was preceded by a phase of multi-spectral image capture with a UAV. Stage 3 (crop vigor monitoring) involved the final monitoring, which included the delineation of homogeneous zones for smart sampling, multi-spectral drone monitoring, NDVI monitoring with an HMA sensor, and vegetative sampling. Finally, Stage 4 (data analysis) comprised data analysis, where normality and significance tests were performed, along with the creation of a correlation matrix to assess the relationships between variables.

### 2.1. Experiment Installation

The experiment took place between 30 October 2024 and 27 March 2025 on a 1.50 ha non-irrigated area at Herdade da Comenda—Innovation Centre of the National Institute for Agricultural and Veterinary Research in Elvas, Alentejo region, Portugal, with local coordinates 38.894° N, −7.055° W (Figure 2).

According to the climatic classifications of Köppen, [15], the region’s climate is defined by the Csa Mediterranean climate. Specific information for the Elvas meteorological station, covering monthly and annual values of the main climatic elements, with an emphasis on average air temperature (Tm; °C) and precipitation (P), are shown in Table 1.

According to the FAO classification, the soil in the experimental area is Luvissoil, which corresponds to the Mediterranean soils Pag and Sr [17].

The experimental area was established with annual ryegrass (*Lolium multiflorum* L.) following a no-till itinerary. The annual ryegrass crop was sown with an inter-row of 0.18 m and a depth of 0.01 m. The sowing doses were 1400 seeds m^−2^ sown on 30 October 2024. Before sowing, on 28 October 2024, sufficient doses of phosphorus (16.1 kg ha^−1^ P_2_O_5_) and potassium (13.8 kg ha^−1^ K_2_O) were applied, along with an evenly distributed dose of N (27.6 kg ha^−1^ N) to ensure the correct emergence of the crop. Weed and disease control was carried out following good experimental practices.

### 2.2. Top-Dressing N Fertilization at a Variable Rate

To carry out this experiment, on 5 March 2025, the OTG sensor was installed on a New Holland 140 Dynamic Command tractor, positioned at a height of 2.85 m from the soil surface (Figure 3). This sensor was installed in the upper part of the tractor cab, in equipment compatible with the ISOBUS protocol.

After installing the OTG sensor on the agricultural machine, the sensor was calibrated. The sensor’s field of view was determined to be 28.5 m, corresponding to ten times the height of its installation. The systems operate as a passive sensor, relying on natural sunlight for its operation. To ensure data accuracy under varying light conditions throughout the day, the sensor is equipped with two additional sensors that measure both the light intensity and the sun angle. This allows the system to continuously correct for fluctuations in ambient light, ensuring reliable and consistent data collection. The system operates by capturing images of the vegetation in real time. These images are then processed by integrated AI algorithms that serve a crucial function beyond mere analysis; they clean the captured data by identifying and eliminating non-vegetative elements such as pathways, stones, and areas with no vegetation. This pre-processing step ensures that the resulting agronomic prescriptions are based solely on relevant crop information, which are then immediately and precisely applied by the fertilizer unit.

Real-time variable rate fertilization in the area under evaluation was carried out on 6 March 2025, following the sensor’s instructions. Prior to the operation, the wi-fi sensor–machine connection was established, after which it was possible to access the field operation platform, on which the data relevant to the experiment was filled in, including the implement setting; application width; field information; date; and choice of operation of interest, specifically real-time variable rate fertilizer application. To configure the real-time VRA operation, the following parameters were determined on the platform (Table 2).

N fertilization was implemented using a commercial ammonium nitrate fertilizer (27% N). Fertilizer application rates were dynamically adjusted to meet crop nutritional demands considering the variability in biomass production potential across the experimental plots. The maximum fertilizer dose was established at 150 kg ha^−1^ of product, consistent with agronomist recommendations based on historical plot data and findings from previous experiments. A dynamic dosing strategy was employed, allowing for real-time adjustments. The minimum dose was set at 50% less than the maximum recommended dose. In areas identified with a high production potential, fertilizer application rates increased to facilitate the maximum yield. Conversely, in low potential areas, application rates were reduced to optimize input efficiency and minimize losses in less productive zones.

### 2.3. Optical Sensors for Monitorization

The UAV multispectral camera was used in this study on 5 March 2025 (Moment I) to monitor the experimental area before top dressing. The multispectral camera has an effective resolution of 2 megapixels, and the images are processed in 4K resolution. This camera is a passive sensor, so it depends on an external light source shining on the vegetation, which in this case was natural sunlight. On 27 March 2025 (Moment II), new images were taken. The orthophotomaps for both dates were generated using Pix4Dfields v.2.7.2 software, in which the collected images were radiometrically calibrated. Finally, the IVs were generated and analyzed using Quantum Geographic Information System (QGIS) software v. 3.40.6 [18].

The HMA optical sensor (Greenseeker^®^, Trimble, CO, USA) is an active sensor and therefore emits light pulses, measuring the light reflected in the red (R: ~660 nm) and near-infrared (NIR: ~770–800 nm) bands on vegetation. In this study, this sensor was used to obtain NDVI values (NDVI_HMA) from 20 different points in the experimental area.

### 2.4. Smart On-Farm Sampling

The sampling points were defined based on the spatial analysis of the fertilizer application rates prescribed by the OTG sensor. The captured images were processed and analyzed using QGIS software v. 3.40.6, making it possible to see the spatial variability in plant biomass in the experimental area (Figure 4). According to the prescription rate ranges, homogeneous zones were created, and points were drawn randomly within these zones to show some representativeness. To adequately represent this variability, 20 georeferenced sampling points were defined. The number of points was considered sufficient to capture the range of variability observed in the OTG NDVI values, optimizing spatial representativeness without compromising the operational feasibility of field collection. Data on PFM, CP, CPyield, NUp, and NDVI_HMA were collected at these 20 points. The respective maps were generated using the Inverse Distance Weighted (IDW) interpolation method.

The field procedure for locating the sampling points involved the precise navigation to each of the 20 georeferenced locations. The coordinates of these points, derived from the spatial analysis of OTG sensor data and visualized in QGIS software v. 3.40.6, were loaded onto a differential GPS (DGPS) receiver Spectra Geospatial MobileMapper 60 (Trimble Inc., Westminster, CO, USA) equipped with Android^TM^ 10, which provided a positional accuracy of approximately 0.5 m. Upon arriving at each designated point, the precise location was visually confirmed against the generated map (Figure 4) and marked with a colored flag to ensure consistency during subsequent data collection events. This rigorous localization process was critical for accurately correlating collected field data with the spatially variable fertilizer application rates and biomass potential identified by the sensor.

#### 2.4.1. Collecting Digital Samples

Digital assessment of crop status was performed through the collection of NDVI data from two primary sources. The primary source was the UAV, which integrates a multispectral sensor camera mounted on the application equipment. This system continuously captured spectral reflectance in the red (R), green (G), blue (B), red-edge (RE), and near-infrared (NIR) bands, allowing for the calculation of real-time NDVI values across the entire delimited AoI. As a complementary method, point-based NDVI readings were acquired using the HMA sensor. The HMA sensor emits its own light and measures reflected light, calculating NDVI values at a specific sampling location. These NDVI_HMA readings were systematically taken at each of the 20 pre-defined georeferenced sampling points to provide discrete, high-resolution measurements of crop vigor at representative locations.

#### 2.4.2. Collecting Vegetal Samples

At the 20 sampling points, the plant material was cut with electric scissors over an area of 1 m^2^, and the PFM produced was weighed to calculate the yield (kg ha^−1^). After weighing the green matter of the samples, they were put in an oven for 24 h at a temperature of 60 °C and weighed again to determine the PDM (kg ha^−1^). Residual moisture was determined in the laboratory using the gravimetric method. After this, the samples were crushed and sieved. To determine the PNC (%) and CP (%), the process used was the Kjeldahl method [19].

In this way, it was possible to calculate the CPyield (kg CP ha^−1^). In this study, NUp was considered by multiplying PDM by PNC as an indicator of the crop’s N nutritional status.

### 2.5. Statistical Analysis

Initially, the data was organized and analyzed using the R v.4.3.2 software, and univariate descriptive analyses were carried out using the psych v. 2.4.3 CRAN Repository package [20,21]. Measures of central tendency and dispersion (mean, median, standard deviation, coefficient of variation) were obtained for the variables PFM, PDM, PNC, CP, CPyield, and NUp and the spectral indices NDVI_OTG, UAV_I, UAV_II, and NDVI_HMA.

The normality of the dependent variables was checked using the Shapiro–Wilk test for each of the continuous variables. In addition, histograms were generated to visualize the distribution of the data and boxplots to identify possible outliers and compare the distribution of the variables according to the treatments.

To test significant differences between treatments, a significance level (α) of 0.05 was used for all analyses. Variables with a non-normal distribution were analyzed using the Kruskal–Wallis test, while those with a normal distribution were analyzed using ANOVA. The correlation coefficients were classified according to their intensity and statistical significance.

To explore the relationships between the spectral indices captured by the sensors and the agronomic indicators of the crops, a bivariate correlation analysis was carried out using Pearson’s correlation coefficient.

## 3. Results

### 3.1. Inferential and Descriptive Statistics

After removing the outliers, the statistical analysis was carried out, and the cleaned data is shown in Table 3.

Based on the data shown in Table 3, which summarizes the results obtained from 19 points sampled in the study area, a descriptive analysis was carried out to understand the variability of the crop’s production, nutritional, and spectral parameters. It was observed that PFM varied between 5800.00 and 20,400.00 kg ha^−1^, with an average of 11,033.33 kg ha^−1^ and a CV of 37.16%, indicating moderate to high variation between sampling points. PDM averaged 2402.87 kg ha^−1^, with minimum and maximum values of 501.54 and 5079.31 kg ha^−1^, respectively, as well as a high CV of 52.40 percent, suggesting significant heterogeneity in the accumulation of dry biomass.

In terms of nutritional composition, the PNC varied from 1.49% to 3.56%, with an average of 2.22% and a CV of 23.89%. CP, both as a percentage and in kg ha^−1^, also varied widely: from 9.31% to 22.23% (average 13.87%) and from 54.72 to 664.92 kg ha^−1^ (average 329.25 kg ha^−1^). CPyield revealed the highest CV compared to CP, with a CV of 50.13%. This range reflects significant variations in the nutritional quality of the forage produced.

NUp showed an average of 52.68 kg ha^−1^, varying between 8.76 and 106.39 kg ha^−1^, and with a CV of 50.13%, which indicates significant variability in the efficiency of N absorption by the plants. As for the spectral indices, the NDVI_OTG had little variation, with values between 0.40 and 0.44 (average of 0.42 and CV of just 2.69%), which suggests homogeneity in the vegetative vigor measured by this index among the field of experiment.

On the other hand, the UAV_I and UAV_II indices showed greater variation, especially UAV_II, with a coefficient of variation of 14.13 percent. The NDVI_HMA index, on the other hand, had an average of 0.72 with a CV of 9.49%, indicating relative homogeneity, although there was still enough variation to justify analyzing it.

To summarize, the parameters with the greatest variation, such as PDM, CPyield, and NUp, stand out for highlighting the spatial heterogeneity of the experimental area. These variations reinforce the importance of adopting site-specific management practices, with a view to greater agronomic efficiency and sustainability of the production system.

### 3.2. Normality Test and Significance Tests

The results of the statistical analysis of the variables studied included the application of the Shapiro–Wilk normality test followed by the choice of the appropriate significance tests depending on the behavior of the data, as presented in Table 4.

The results of the Shapiro–Wilk test indicated that the PFM, PDM, PNC, and CP variables had *p*-values of less than 0.05, which shows that their data does not follow a normal distribution. For these variables, the Kruskal–Wallis test was applied, which does not require normality of the data.

On the other hand, the variables CPyield, NUp, NDVI_OTG, UAV_I, UAV_II, and NDVI_HMA showed *p*-values greater than 0.05, indicating that the data can be considered normally distributed. Therefore, ANOVA was used for these variables. Figure 5 shows the histograms of the variables analyzed in the study, providing a visual representation of the frequency distribution of the data.

The PFM, PDM, PNC, and CP variables showed asymmetrical distributions, with a trend to the right (positive asymmetry), showing that most of the values are concentrated in the lower ranges, with some high extreme values. This confirms the results of the Shapiro–Wilk normality test, which indicated a non-normal distribution for these variables.

The CPyield and NUp variables also showed some asymmetry, albeit less marked. This distribution reinforces the decision to apply non-parametric tests, such as Kruskal–Wallis, to these variables if there is any doubt about the homogeneity of the variance or the shape of the curve.

On the other hand, variables such as NDVI_OTG, UAV_I, UAV_II, and NDVI_HMA showed more symmetrical distributions that were close to normality. The histograms of these variables (Figure 5) showed a relatively uniform dispersion of data around the mean, which is in line with the *p*-values of the Shapiro–Wilk test, which did not reject the hypothesis of normality. This justified the use of ANOVA in these cases.

The results of evaluating the influence of soil spatial variability, applied N dose, and on-the-go sensor readings on crop performance and nutritional parameters, indicating the *p*-values for the influence of “Homogeneous Zone Class”, “Dose N applied”, and NDVI_OTG on each dependent variable, are presented in Table 5.

The analysis revealed that, at conventional levels of statistical significance, the VRA strategy maintained performance without a significant reduction in yield. Specifically, the “Homogenous Zone Class” did not show a statistically significant effect on any measured attribute, including PFM (*p* = 0.291), PDM (*p* = 0.581), PNC (*p* = 0.648), CP (*p* = 0.648), CPyield (*p* = 0.681), NUp (*p* = 0.681), NDVI_OTG (*p* = 0.562), UAV_I (*p* = 0.888), UAV_II (*p* = 0.678), or NDVI_HMA (*p* = 0.335). Furthermore, the NDVI_OTG itself, when considered as an independent predictor, did not show a statistically significant effect on any of the dependent variables, with *p*-values ranging from 0.175 to 0.456, while its self-correlation was not applicable for this test.

Despite this overall lack of statistically significant effects from the tested factors, a particularly noteworthy finding emerged regarding the “Dose N applied”. The high *p*-values for critical productivity parameters like PFM (*p* = 0.515) and PDM (*p* = 0.723) indicate that the variable nitrogen doses applied did not lead to a statistically discernible reduction in these biomass yields.

### 3.3. Correlation Matrix

To understand the relationship between the study’s dependent variables, a correlation matrix was created. The results of the correlation matrix are shown in Figure 6.

Spearman’s correlation matrix analysis showed significant relationships between agronomic variables and spectral indices, with PFM being one of the main indicators of production performance. There was a very strong correlation between PFM and PDM (R^2^ = 0.85), CP (R^2^ = 0.95), and NUp (R^2^ = 0.95), indicating that the increase in biomass is directly associated with an increase in nutritional quality and increased nutrient absorption efficiency. The NUp variable also showed a strong correlation with PDM (R^2^ = 0.86) and moderate correlations with UAV_II (R^2^ = 0.55) and NDVI_HMA (R^2^ = 0.53). About spectral indices, UAV_II stood out with a moderate correlation with CP (R^2^ = 0.58) and a strong correlation with the NDVI_HMA parameter (R^2^ = 0.70), demonstrating its complementary applicability in the remote monitoring of plant N nutrition attributes. Also, this suggests that these indexes were sensitive to the variation in N absorption and accumulation induced by variable rate fertilization.

The correlation matrix indicates that NDVI_OTG had a positive and moderate linear correlation with important agronomic variables, such as PFM (R^2^ = 0.52).

### 3.4. Maps of Sampling Points

The maps of the spatial distribution of PFM and CP of the field cultivated with ryegrass and subjected to variable rate N fertilization are shown in Figure 7. In image (a), the spatial variability of PFM (kg ha^−1^) can be seen, highlighting areas of greater biomass accumulation located predominantly in the central and southern regions of the experimental field.

The areas with the darkest shades represent the highest yield values, above 18,512.54 kg ha^−1^, while the lighter areas indicate lower yield performance, below 7467.67 kg ha^−1^. Image (b) shows the spatial distribution of CP content (%), with a higher concentration of CP in the north and southwest of the experimental area, as shown by the darker colors. The contents varied between 20.79% and 10.81%, reflecting the effect of N fertilization on the nutritional quality of the forage. The joint analysis of the images indicates a positive correlation between the zones with the highest productivity and the highest protein levels, suggesting that the crop’s response to N fertilization is spatially dependent, which reinforces the importance of using PA technologies for the efficient management of inputs.

Figure 8 shows the spatial distribution maps of CPyield productivity and NUp in the forage subjected to variable rate N fertilization.

Image (a) shows the spatial variability of CPyield yield (kg ha^−1^), with higher values (591.31 to 517.62 kg ha^−1^) concentrated in the central and southeastern regions of the experimental area, as shown by the darker shades. The lighter regions, located in the north and parts of the south of the area, indicate lower protein yields, with values of less than 124.59 kg ha^−1^. The image (b) shows the spatial distribution of NUp, expressed in kg of crude protein per kg of N applied. The areas with the highest efficiency (9.61 to 8.63 kg^−1^) are concentrated mainly in the center and southwest of the plot, while the lowest efficiency values (<1.93 kg^−1^) occur on the northern and southern edges of the area. The comparison between the maps shows that the areas with the highest protein productivity do not always coincide with those with the highest N use efficiency, which highlights the importance of integrated spatial analysis for rational fertilizer management, seeking to optimize production with less environmental impact and greater sustainability of the forage system.

Figure 9 presents the spatial distribution map of the NDVI values obtained using the HMA optical sensor.

The NDVI_HMA values ranged from 0.60 to 0.82, reflecting differences in the forage’s vegetative vigor in response to variable rate N fertilization. Areas with higher NDVI_HMA values (0.82 to 0.76), shown by the darker shades, are concentrated in the central and northeastern parts of the experimental plot. This indicates greater biomass density and photosynthetic activity. Conversely, areas with lower values (below 0.67), highlighted by orange and yellowish colors, are mainly located in the southern and southeastern regions, suggesting poorer plant development.

### 3.5. Maps of Homogeneous Zones and Fertilizer Application

Figure 10 illustrates the spatial distribution of homogeneous zones for N fertilization rates, derived from the prescriptions generated by the OTG sensor. These zones were delineated based on the observed spatial variability of crop vigor within the experimental area, allowing for the optimization of nutrient input application by grouping regions with similar agronomic characteristics.

The prescribed fertilizer rates, expressed as relative values, ranged from 0.5 to 0.9 and were categorized into four distinct classes (Figure 10). Areas designated with higher relative rates (0.8 to 0.9), visually represented by darker shades, were predominantly concentrated in the southern sector of the experimental plot. This spatial pattern suggests regions with potentially greater nutritional demand or higher production potential. Conversely, zones with lower relative rates (0.5 to 0.6), depicted in lighter tones, were more fragmented and dispersed throughout the area, indicating a reduced need for N fertilization.

To further detail the VRA, Table 6 presents the total fertilizer doses applied per homogeneous zone, alongside a comparison with a hypothetical fixed-rate application scenario.

Table 5 details the area, percentage representativeness, prescribed fertilizer dose per hectare, and total fertilizer consumed for each homogeneous zone under the VRA strategy compared to a fixed-rate control. The hypothetical fixed rate row represents a scenario where the entire 1.5 ha plot received a uniform application of 150 kg ha^−1^ of fertilizer, resulting in a total consumption of 225 kg.

Under the VRA, the plot was divided into four classes (1, 2, 3, and 4) based on the OTG sensor prescription. These classes represent varying levels of nutritional demand, with corresponding fertilizer doses ranging from 90 kg ha^−1^ (Class 1) to 135 kg ha^−1^ (Class 4). Although Classes 1 and 2 represent very small areas (0.002 ha and 0.017 ha, respectively), Classes 3 and 4 collectively covered most of the field (approximately 50% each).

The total amount of fertilizer applied with the VRA strategy was calculated by summing the consumption across all classes, resulting in 0.180 kg (Class 1), plus 1.785 kg (Class 2), plus 88.920 kg (Class 3), and plus 99.765 kg (Class 4), equal to 190.65 kg. This demonstrates a substantial reduction in total fertilizer use compared to the fixed-rate application. Specifically, the variable rate strategy resulted in a savings of 34.35 kg of fertilizer over the 1.5 ha field, which translates to a savings of 22.9 kg of fertilizer per hectare. Considering the fixed rate of 150 kg ha^−1^ as the recommended dose, the use of the OTG sensor allowed for an approximate 15.23% reduction in the total amount of fertilizer applied.

## 4. Discussion

### 4.1. Optical Sensors in Nitrogen Fertilization Management Sensors’ Response

The observed spatial variability in key agronomic parameters, such as PFM, PDM, CPyield, and NUp, underscores the inherent heterogeneity of the experimental area. PFM and PDM exhibited moderate to high coefficients of variation (37.16% and 52.40%, respectively). This heterogeneity reinforces the premise that uniform fertilizer applications may lead to suboptimal nutrient use efficiency and uneven crop performance, in accordance with [22].

The study employed two distinct optical sensors: the on-the-go system and the handheld multispectral active optical sensor. Interestingly, the NDVI_OTG values showed very low variation (CV of 2.69%), suggesting a relatively homogeneous vegetative vigor as measured by this specific sensor across the experimental field. This contrasts with the significant variability observed in actual biomass produced and NUp parameters. This discrepancy might be attributed to the scale of measurement (continuous field-level data vs. point sampling), the specific sensing characteristics of the OTG system, or a potential saturation effect of NDVI at higher biomass levels, where small differences in vigor are not accurately reflected by the index. Arjasakusuma et al. [23] demonstrated that differences in NDVI values can arise from the resolution and scale of measurement, and continuous field-level data provide a comprehensive view of vegetation health, while point sampling may miss spatial variability and lead to discrepancies. Crusiol et al. [24] also pointed out that the characteristics of the sensing system, such as sensor height, time of day, and environmental conditions, can significantly impact NDVI readings. For instance, NDVI values tend to decrease with higher sensor heights, and readings are most reliable at specific times of the day. However, it should be noted that the OTG sensor is a passive sensor, but it is calibrated in real time to correct its readings according to the intensity and angle of incidence of sunlight. This fact, added to its ability to select only vegetation as a target of interest through AI, excluding noise that may influence readings, is significant in reducing the CV of the indices read. Further investigation into the specific algorithms and spectral bands utilized by the OTG system would be beneficial to fully understand its sensitivity under these conditions.

The HMA sensor, while also showing relative homogeneity (CV of 9.49%), demonstrated sufficient variation to justify its analysis, and its NDVI values correlated positively and moderately with PFM (R^2^ = 0.52). Moreover, the UAV_II index exhibited strong correlations with both CP (R^2^ = 0.58) and the NDVI_HMA parameter (R^2^ = 0.70) and moderate correlation with NUp (R^2^ = 0.55). This suggests that multispectral data collected by UAV at the final monitoring stage, particularly, and NDVI_HMA are more sensitive indicators of N status and biomass accumulation under the study conditions, responding to variations induced by the variable rate fertilization. The higher variability in UAV_II (CV of 14.13%) compared to NDVI_OTG further supports its utility in detecting subtle changes related to N absorption and accumulation.

Overall, while the OTG sensor effectively delineated homogeneous zones for variable application, the ground-truthing data and the HMA, along with UAV data, provided a more nuanced representation of the physiological and nutritional variability within the crop. This highlights the complementary nature of continuous, platform-based sensing and targeted, handheld measurements for comprehensive crop assessment. These findings are also highlighted by Wang et al. [25], who said that integrating continuous, platform-based sensing with targeted, handheld measurements creates a robust framework for comprehensive crop assessment. This synergy enhances data accuracy, spatial coverage, and temporal resolution, ultimately supporting more precise and sustainable agricultural practices.

It is necessary to consider that the density of verification data in the field may have limitations in capturing all spatial variability and validating the finer details of the spatial maps. While the 19 sampling points were strategically distributed to represent the homogeneous zones delineated by the sensor, it is acknowledged that for a heterogeneous plot of 1.5 ha, this sample size may not be sufficient to fully capture all spatial variability. However, the study’s reliance on continuous OTG sensor data serves to mitigate this limitation. The high-resolution data stream from the OTG sensor provided a comprehensive, sub-field-scale representation of crop vigor that would have been impossible to achieve with point sampling alone. This highlights the complementary nature of the two methodologies: point sampling provided crucial validation for key parameters, while the continuous sensing captured the full extent of the spatial heterogeneity. For future research, a higher-density ground-truthing grid or the use of a more advanced stratified sampling approach based on sensor maps would be beneficial to provide even more robust validation of the sensor-derived prescriptions.

To provide a clearer understanding of the technical differences and operational requirements of each system, a summary of their key specifications, including data acquisition and processing, is presented in Table 7.

The data presented in Table 7 highlights the operational trade-offs and complementary benefits of each sensing platform. The UAV system, while providing a high spatial resolution (5.3 cm/pixel) and valuable multi-band data, generated a substantial digitization footprint of 3 GB per hectare, with an associated processing time of approximately two hours. This large data volume and processing effort can be a limiting factor for real-time or large-scale applications. In contrast, the OTG sensor generated a significantly smaller footprint of 106 Mb per hectare, and its data processing was instantaneous due to its on-board AI. This makes the OTG system highly efficient for practical, on-the-go applications. The HMA sensor, with its point-based sampling, had a negligible data footprint (98 Kb) but required interpolation and manual effort, making it suitable for targeted ground-truthing rather than continuous mapping. The fusion of data from these diverse sensors is, therefore, a robust approach that leverages the strengths of each platform—the OTG for real-time application, the UAV for high-resolution insights, and the HMA for validation—while managing their respective operational limitations.

These results corroborate several well-documented advantages and disadvantages of each sensor type. Handheld sensors are highly portable and easy to use, making them suitable for quick, on-site nutrient assessment [26,27]. However, their primary limitation lies in their operational efficiency, as collecting data is both time-consuming and labor-intensive, especially across large fields. Furthermore, as these sensors provide only point measurements, they may fail to capture the full spatial variability of a field [26].

In contrast, Unmanned Aerial Vehicles (UAVs) equipped with multispectral sensors can capture high-resolution data over large areas, enabling precise and efficient monitoring of crop health and nutrient status [28]. The automation of data collection significantly reduces the labor and time required compared to manual methods [26]. Despite these benefits, UAV systems and their sensors can be expensive and require a high level of technical expertise for both operation and data processing [29]. Additionally, the data can be affected by variations in solar illumination and atmospheric conditions, which often necessitates frequent calibration for accurate readings [30].

Finally, on-the-go sensors, like the OTG system used in this study, provide a unique advantage by delivering real-time data on nutrient variability, allowing for immediate adjustments in fertilizer application. These sensors are particularly efficient for large-scale farming operations as they can cover extensive areas quickly. A key consideration for their effectiveness is the distance from the crop canopy, which must be carefully optimized for accurate readings. It is also important to note that integrating on-the-go sensor data with other data sources, such as UAV imagery, can be complex and may require advanced data fusion techniques to achieve optimal results [31]. The low CV observed in our NDVI_OTG data, which contrasts with the significant variability in actual biomass, can be directly attributed to the technical advantages of our AI-driven system. Traditional vegetation screening methods, such as fixed-threshold algorithms, are prone to including noise from shadows, soil, and mixed pixels, which can artificially inflate data variability [32,33]. While these methods are easy to implement, they often struggle with the complex and changing conditions of an agricultural field. As pointed out by some authors [34,35], on-the-go AI vegetation screening algorithms offer significant advantages over traditional manual methods in efficiency, accuracy, and cost-effectiveness. Traditional approaches, such as manual inspections or expert photo interpretation, are labor-intensive, time-consuming, and can be subjective. These methods are often limited by the complexity of vegetation backgrounds and are prone to human bias, making them less accurate than automated solutions [35].

In contrast, the on-the-go AI algorithm of our sensor leverages advanced computer vision to filter out non-vegetative elements in real time. This capability provides a distinct advantage over both traditional thresholding and other machine learning approaches. For example, studies have shown that deep learning models can achieve vegetation classification accuracies of over 95%, while older algorithms like Random Forest often fall short, with a mean intersection over union (MIoU) of around 0.47 and an F1-score of 0.55 [36]. By using a more sophisticated AI model, our sensor ensured that the NDVI readings were based on a much cleaner dataset, leading to a more accurate representation of the crop’s true vigor and a lower, more reliable CV. The integration of advanced computer vision and machine learning techniques enabled real-time data processing for large-scale vegetation monitoring, reducing human effort and providing a more reliable alternative for sustainable and resilient agricultural management. This high degree of accuracy and noise removal allowed this OTG sensor-based approach to effectively delineate homogeneous zones for variable application, providing a more robust basis for precision nitrogen management compared to what would be possible with simpler screening methods.

### 4.2. Efficiency and Reduction in Fertilizer Used

One of the most significant findings of this study is the demonstrable efficiency gained through the implementation of variable rate N fertilization. By dynamically adjusting fertilizer application based on the OTG sensor prescription, a total of 190.65 kg of fertilizer was applied to the 1.5 ha plot, compared to a hypothetical 225 kg if a fixed rate of 150 kg ha^−1^ had been uniformly applied. This translates to a substantial 15.23% reduction in total fertilizer applied or 22.9 kg of fertilizer saved per hectare.

This reduction in N input is highly significant for both economic and environmental sustainability. From an economic perspective, reducing fertilizer consumption directly lowers input costs for farmers, contributing to improved profitability. Environmentally, minimizing excess N application mitigates potential negative impacts such as nitrate leaching into groundwater, nitrous oxide emissions (a potent greenhouse gas), and eutrophication of surface waters as referred to by Govindasamy et al. [37] and Zhao et al. [38]. The savings observed are in line with the reductions recorded by Shi et al. [39] and Chen et al. [40] of between 20 and 40% of the N applied to improve NUE and crop yield. In this precision N management of various crops, Castillo-Díaz et al. [41] adds that these levels of savings resulted in better economic outcomes and sustainability.

Assuming that the test was carried out on a 1.5-hectare plot, it should be borne in mind that the system is constantly being recalibrated by the AI it incorporates. However, the difficulty encountered in this test due to the small size of the plot under study must be seen in the context of the average size of farmers’ plots. The average size of agricultural plots varies significantly across different regions and farming systems. There are more than 570 million farms worldwide, with most being small and family-operated. Small farms (less than 2 hectares) operate about 12% of the world’s agricultural land, while family farms manage about 75% [42]. This indicates a prevalence of small-scale farming globally. In central and eastern Europe, traditional land-use systems had average plot sizes of several hectares, but modern plots can reach sizes of 200 hectares or more [43]. In contrast, in the United States, the mean and median field sizes are 19.3 ha and 27.8 ha, respectively, with a significant number of fields being smaller than 36.1 ha [44]. Larger plots tend to benefit more from continuous farming patterns, which enhance productivity [45]. However, smaller plots are often preferred for their aesthetic and ecological benefits, as seen in European landscapes where smaller plots are associated with higher visual preferences [43]. The Mediterranean region is characterized by highly fragmented agricultural land. This fragmentation is a result of traditional land inheritance and transmission systems, leading to small production units often extending over less than 2 hectares. This fragmentation poses economic and environmental challenges that need to be addressed through targeted policies and management strategies [46]. The strategy of differentiating fertilizer rates based on homogeneous zones of production potential, as defined by the OTG sensor, proved effective. While Classes 1 and 2 received lower doses and comprised smaller areas, most of the field (Classes 3 and 4) received intermediate to higher doses, optimizing nutrient supply to the most productive areas while avoiding over-fertilization in less productive regions. This site-specific approach is crucial for achieving a high NUE, which is a key goal in sustainable agriculture.

### 4.3. Practical Implications of Sensor Use in Winter Fodder Crops in Mediterranean Climate

The results of this study have substantial practical implications for the management of winter fodder crops in Mediterranean climate. These regions are often characterized by heterogeneous soil conditions, variable rainfall patterns, and diverse farming practices, making uniform fertilizer application less efficient. The successful deployment of the OTG sensor for delineating homogeneous zones and implementing VRA demonstrates a viable strategy for precision N management in this specific context.

The ability to dynamically adjust fertilizer rates allows farmers to respond to real-time crop demands, which is particularly critical in fodder crops where biomass production and nutritional quality (e.g., CP content) are paramount. The spatial maps of PFM, CP, CPyield, and NUp clearly illustrate that crop response to N fertilization is highly spatially dependent. Areas with higher productivity and protein levels often coincided, but not always with the highest NUE. This highlights the complex interaction between N application, crop growth, and nutrient assimilation, emphasizing the need for integrated spatial analysis for rational fertilizer management.

The ease of integration of the OTG sensor with existing machinery allows for on-the-go adjustments, reducing the need for extensive pre-field mapping or post-application analysis for simple adjustments. This operational feasibility is a key factor for farmer adoption. The observed reduction in fertilizer use, combined with the potential for optimized yields and improved forage quality, provides a strong economic incentive for implementing sensor-based N fertilization in these cropping systems. Furthermore, the capacity to identify and manage less productive areas separately can reduce the loss of expensive inputs in non-responsive zones, thereby enhancing overall resource efficiency and environmental stewardship.

The economic viability of our proposed VRA strategy is a key factor for its widespread adoption. To highlight its promotion value, we compared the single-operation cost of the OTG system with the cost of fertilizer waste from traditional fertilization. Based on public data, the price of nitrogen-based fertilizers is approximately EUR 341 tonne^–1^ [47]. In this context, a traditional uniform fertilization approach can result in significant financial waste due to nutrient losses and suboptimal application.

The adoption of VRA technology presents a clear economic advantage. The annual cost of variable rate fertilization (VRF) equipment, including depreciation, varies from USD 13 to USD 131 per hectare depending on the farm’s size [48]. This investment is rapidly offset by direct savings. For instance, studies on similar VRF technology have shown a fertilizer cost reduction of USD 13.43 per hectare (95.28 RMB/hm^2^) compared to traditional methods [49]. Our study, which resulted in a 15.23% reduction in total fertilizer use, corresponding to USD 9.12 per hectare, demonstrates a direct saving that can easily justify the initial investment. Furthermore, the economic benefits extend beyond cost reduction. The optimized application of nutrients through VRA has been shown to increase yield profits by as much as USD 142.34 per hectare (1007.62 RMB/hm^2^) in some cropping systems [49]. This strong economic incentive, driven by both cost savings and increased profitability, is a primary driver for promoting the adoption of this technology.

### 4.4. Use of On-the-Go Sensors and Modelling N Fertilization of Fodder Crops in Mediterranean Climate

The application of OTG sensors, such as the system used in this study, represents a significant advancement in the precision management of N fertilization for fodder crops in Mediterranean climates. These sensors provide real-time data on crop vigor, enabling immediate adjustments to fertilizer application rates, which is superior to traditional methods relying on static soil tests or visual assessments. The continuous data stream allows for a highly granular approach to N management, addressing variability at a sub-field scale that would otherwise be overlooked. Once the possible improvements identified in the previous subchapter have been achieved, these on-the-go systems can unlock automation and digitalization in agricultural operations in the Mediterranean environment. While this study successfully demonstrated the practical benefits of on-the-go sensing in a specific Mediterranean climate, these results have theoretical extensibility and provide a strong foundation for comparison with other regions.

The challenges posed by the Mediterranean climate, such as unpredictable rainfall and periods of drought, require a responsive and adaptive approach to N management. A key area for future research would be to explore the universality of these sensor-based strategies by applying them in different climatic regions, such as temperate or tropical climates where environmental variables, like rainfall patterns, are more stable or predictable. Such comparative research would help to isolate the specific impacts of climate on sensor responses and refine the algorithms for broader applicability. For instance, in a temperate region with more consistent soil moisture, the sensor’s readings may be less affected by water stress, allowing for a more direct focus on N uptake.

Current models often rely on generalized crop NUp curves and environmental parameters. Furthermore, exploring the potential of combining on-the-go sensor data with more sophisticated N fertilization models remains crucial. By incorporating real-time sensor data, particularly indices like UAV_II and NDVI_HMA, which showed strong correlations with NUp, these models could be refined to provide more accurate and site-specific N recommendations. Developing dynamic models that integrate sensor outputs, weather forecasts (especially crucial in Mediterranean climates with variable rainfall), and crop growth stage could further enhance the precision and responsiveness of N management.

Furthermore, exploring the potential of combining on-the-go sensor data with remote sensing imagery (e.g., satellite or drone-based) could provide a multi-temporal and multi-scale perspective on crop N status. This would allow for better pre-season planning and in-season adjustments, optimizing fertilizer strategies throughout the crop cycle. The findings from this study lay a strong foundation for the development of such integrated decision-support systems, paving the way for even more sustainable and productive fodder crop systems in Mediterranean agricultural landscapes.

## 5. Conclusions

Firstly, this study effectively validated a cutting-edge OTG sensor for real-time variable rate N application. Its capacity to delineate homogeneous zones based on perceived crop vigor allowed for a dynamic adjustment of fertilizer doses, moving beyond conventional uniform application. While the NDVI_OTG showed less variation than other indices, its utility in driving the VRA strategy was clearly demonstrated through the resulting spatial distribution of applied N.

Secondly, an integrated and complementary approach using different sensors significantly enhanced the understanding of crop response to variable N fertilization. The HMA sensor and particularly the NDVI collected by UAV at final crop monitoring proved to be more sensitive indicators of the crop’s physiological and nutritional status, showing stronger correlations with critical agronomic parameters such as NUp and CPyield. This highlights the value of combining continuous on-the-go sensing with targeted, high-resolution point measurements for comprehensive crop assessment and validation. The spatial maps generated from these parameters further elucidated the complex heterogeneity of crop response across the field.

Thirdly, the research clearly demonstrated significant efficiency gains and reduction in fertilizer use, directly contributing to both sustainability and productivity. By precisely tailoring fertilizer application to site-specific crop needs, this approach not only achieves substantial economic and environmental gains, as evidenced by the notable reduction in total fertilizer applied, but also sets a new precedent for sustainable nutrient management. This study’s success demonstrates that precision agriculture technologies can move beyond theoretical models to deliver tangible benefits, such as enhanced NUE, without compromising productivity. These findings provide a compelling case for the widespread adoption of such technologies, offering a scalable pathway for producers to improve profitability.

Finally, the study was successfully conducted within the specific context of the Alentejo region and its winter fodder crops, validating the robustness and applicability of this technology under Mediterranean conditions. The inherent spatial variability observed in the field, characteristic of such environments, underscores the critical need for precision agriculture technologies. The results confirm that sensor-based variable rate N fertilization is a highly promising and practical solution for improving resource management and environmental stewardship in this unique agricultural landscape.

In conclusion, the integration of advanced optical sensors, particularly the OTG system for application control and complementary HMA sensors for detailed assessment, offers a powerful paradigm for sustainable N management in winter fodder crops. This approach not only optimizes input use and enhances economic viability but also contributes significantly to the environmental sustainability of agricultural practices in Mediterranean climates.

## Figures and Tables

**Figure 1 sensors-25-05811-f001:**
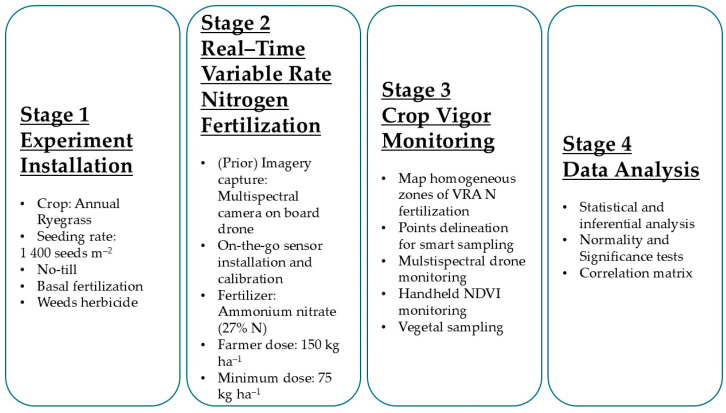
Principal stages of the experiment field.

**Figure 2 sensors-25-05811-f002:**
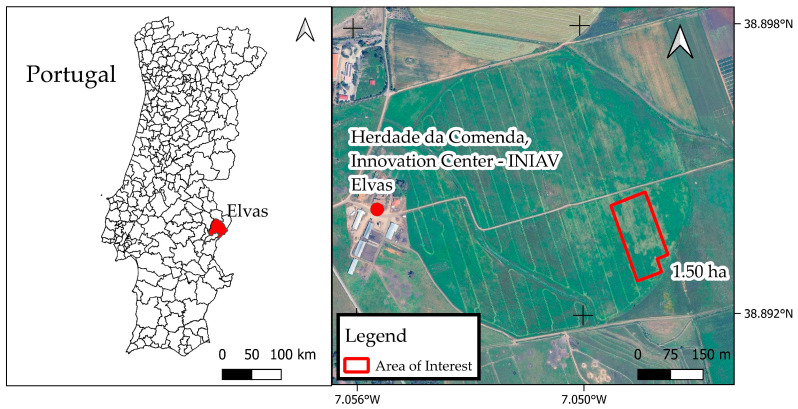
Identification and delimitation of experimental area.

**Figure 3 sensors-25-05811-f003:**
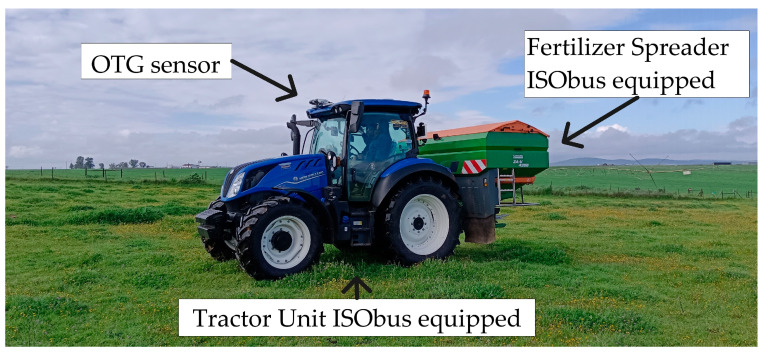
Tractor-machine operator set equipped with ISObus and OTG sensor installed on top of cab.

**Figure 4 sensors-25-05811-f004:**
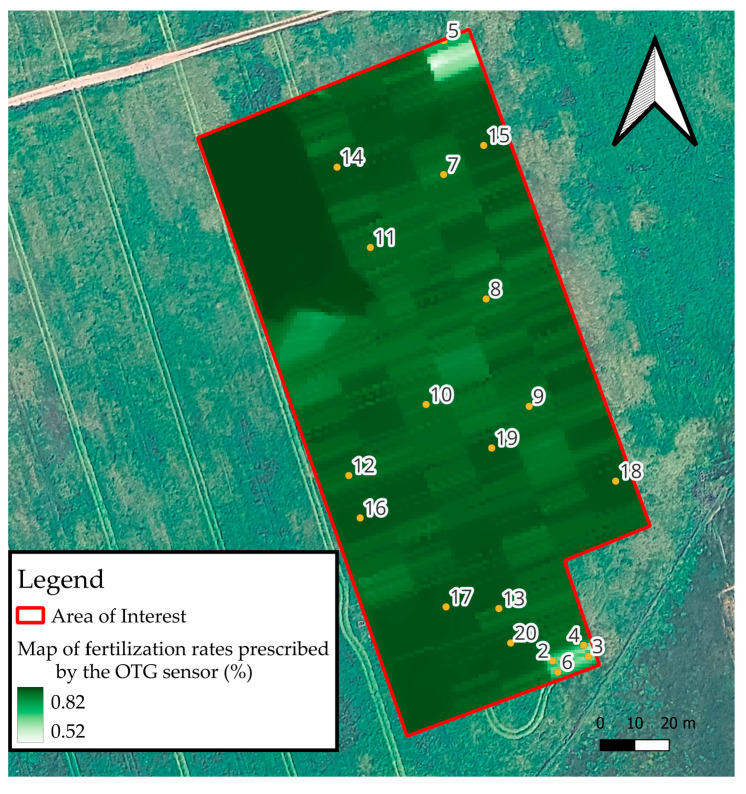
Localization of sampling points according to prescribed taxes of OTG sensor.

**Figure 5 sensors-25-05811-f005:**
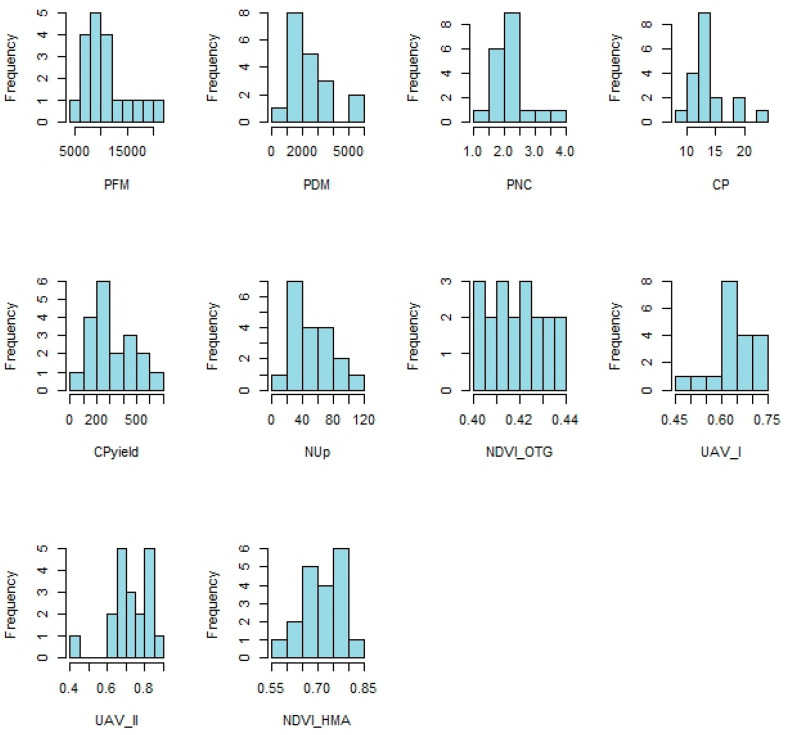
Histograms of analyzed variables.

**Figure 6 sensors-25-05811-f006:**
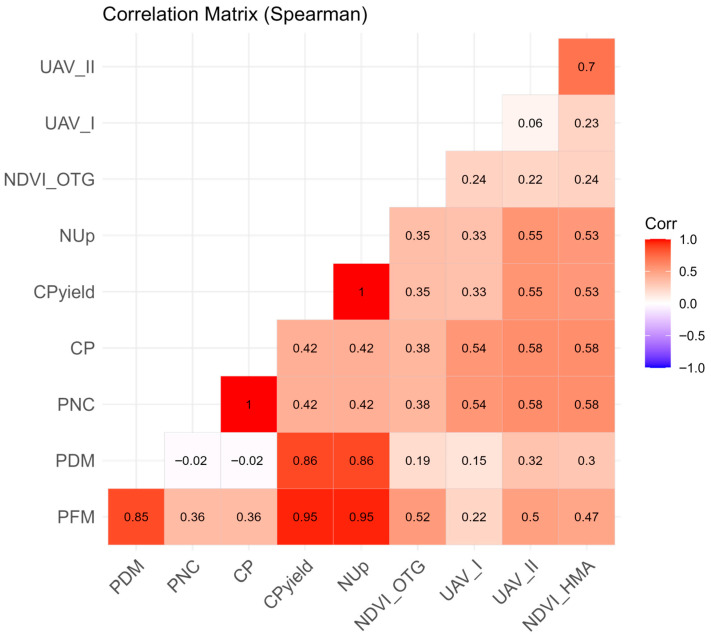
Correlation matrix.

**Figure 7 sensors-25-05811-f007:**
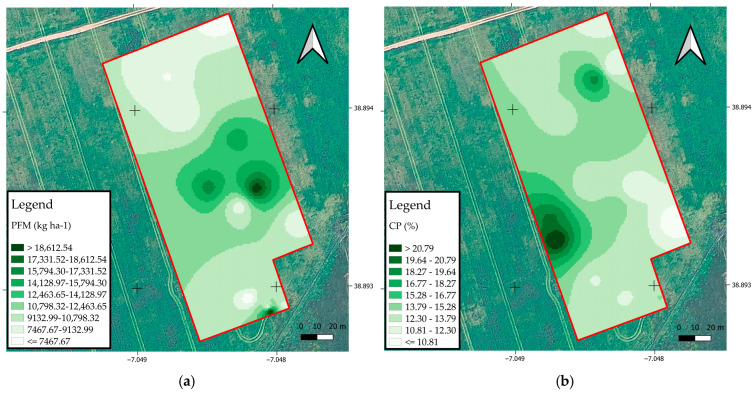
(**a**) Spatial distribution of PFM (kg ha^−1^); (**b**) spatial distribution of CP content (%) of the fodder crop.

**Figure 8 sensors-25-05811-f008:**
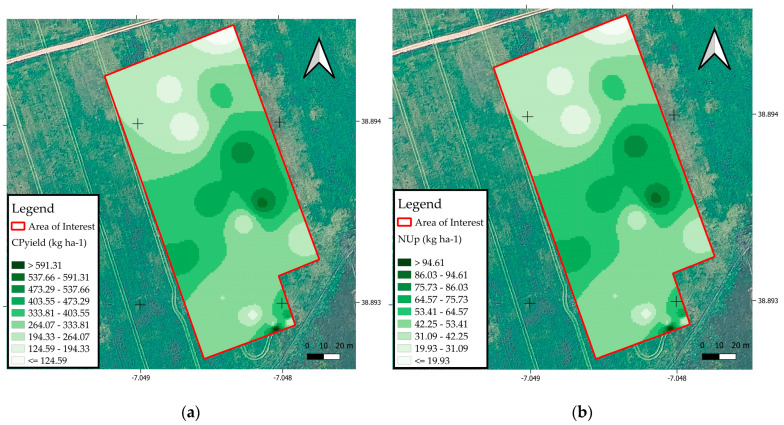
(**a**) Spatial distribution of CPyield (kg ha^−1^); (**b**) spatial distribution of NUp (kg ha^−1^).

**Figure 9 sensors-25-05811-f009:**
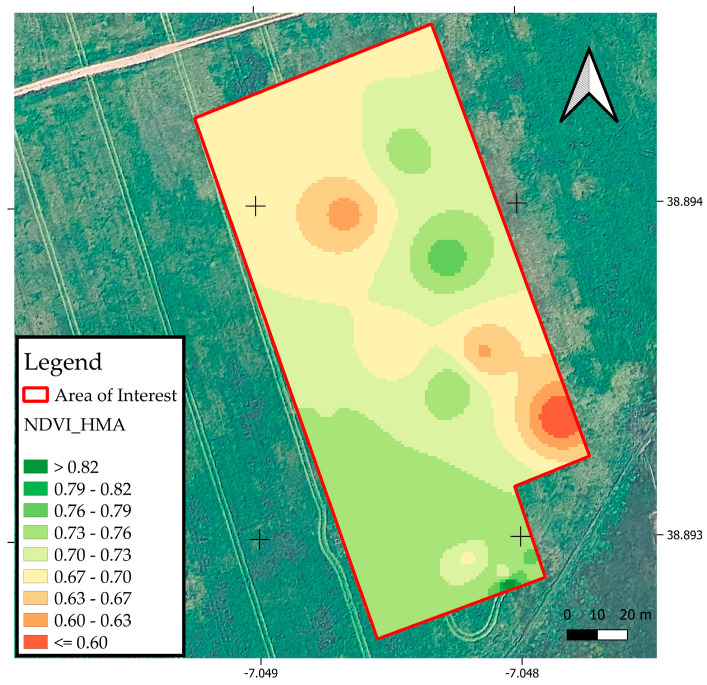
Spatial distribution of NDVI_HMA values.

**Figure 10 sensors-25-05811-f010:**
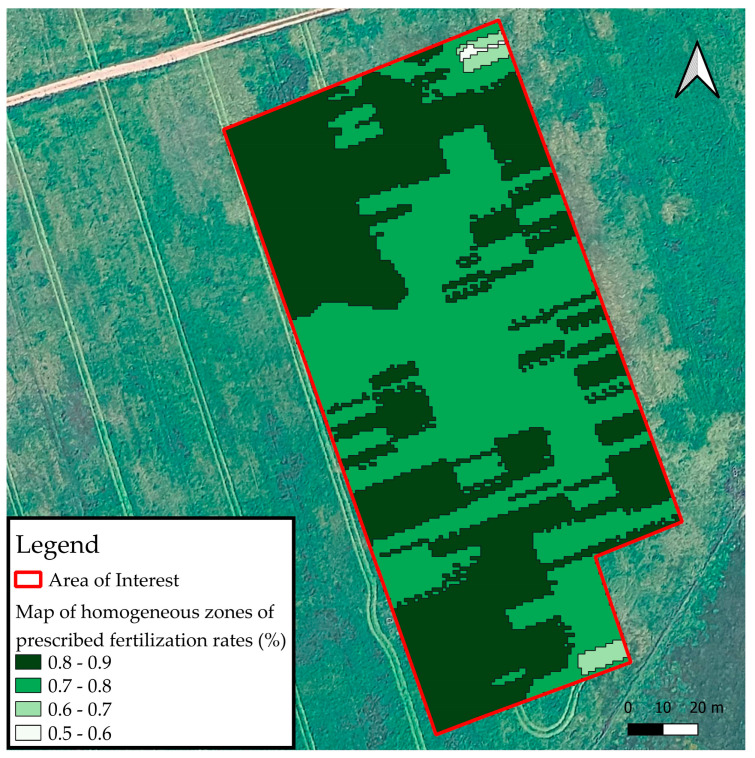
Map of homogeneous zones of fertilization rates prescribed by OTG sensor.

**Table 1 sensors-25-05811-t001:** Climatological normal for values of Tm and P in Elvas [16].

Month	Tm Normal (°C)	P Normal (mm)
October	17.4	58.6
November	12.5	75.1
December	9.7	92.6
January	8.6	63.1
February	10.2	54.6
March	12.3	39.6
April	14.1	51.2

Tm—Average air temperature (°C); P—Precipitation (mm).

**Table 2 sensors-25-05811-t002:** Values of each parameter introduced in the system before the operation.

Parameter	Value
Operation	Variable rate fertilization in real time
Crop	Pasture
Working height	28 m
Fertilizer	Nitro 27–0–0 (27% N)
Maximum dose applied	150 kg ha^−1^
Minimum dose applied	75 kg ha^−1^

**Table 3 sensors-25-05811-t003:** Descriptive and inferential statistical analysis.

	PFM (kg ha^−1^)	PDM (kg ha^−1^)	PNC (%)	CP (%)	CPyield (kg ha^−1^)	NUp (kg ha^−1^)	NDVI_OTG	UAV_I	UAV_II	NDVI_HMA
No. samples	19	19	19	19	19	19	19	19	19	19
Min	5800	502	1.49	9.31	54.7	8.8	0.40	0.50	0.44	0.57
Max	20,400	5079	3.56	22.23	664.9	106.4	0.44	0.72	0.85	0.85
Mean	11,033	2403	2.22	13.87	329.3	52.7	0.42	0.65	0.72	0.72
Median	10,000	2130	2.12	13.23	279.7	44.8	0.42	0.64	0.72	0.74
SD	4100	1259	0.53	3.31	165.1	26.4	0.01	0.05	0.10	0.07
CV	37.2	52.4	23. 9	23. 9	50.1	50.1	2.7	8.0	14.1	9.5

PFM—plant fresh matter (kh ha^−1^); PDM—plant dry matter (kg ha^−1^); PNC—plant nitrogen content (%); CP—crude protein (%); CPyield—crude protein per unit of area (kg ha^−1^); NUp—nitrogen uptake (kg ha^−1^); NDVI_OTG—NDVI of on-the-go sensor; UAV_I—NDVI of unmanned aircraft vehicle at first moment; UAV_II—NDVI of unmanned aircraft vehicle at second moment; NDVI_HMA—NDVI of handheld multispectral active sensor; SD—standard deviation; CV—coefficient of variation.

**Table 4 sensors-25-05811-t004:** Normality test results and best-fitted significance test identified.

	PFM (kg ha^−1^)	PDM (kg ha^−1^)	PNC (%)	CP (%)	CPyield (kg ha^−1^)	NUp (kg ha^−1^)	NDVI_OTG	UAV_I	UAV_II	NDVI_HMA
Shapiro–Wilk (*p*-value)	0.013	0.035	0.007	0.007	0.478	0.478	0.263	0.053	0.045	0.563
Significance test	Kruskal–Wallis	Kruskal–Wallis	Kruskal–Wallis	Kruskal–Wallis	ANOVA	ANOVA	ANOVA	ANOVA	Kruskal–Wallis	ANOVA

PFM—plant fresh matter (kh ha^–1^); PDM—plant dry matter (kg ha^–1^); PNC—plant nitrogen content (%); CP—crude protein (%); CPyield—crude protein per unit of area (kg ha^–1^); NUp—nitrogen uptake (kg ha^–1^); NDVI_OTG—NDVI of on-the-go sensor; UAV_I—NDVI of unmanned aircraft vehicle at first moment; UAV_II—NDVI of unmanned aircraft vehicle at second moment; NDVI_HMA—NDVI of handheld multispectral active sensor.

**Table 5 sensors-25-05811-t005:** *p*–values resulting from significance tests.

Dependent Variables	Homogenous Zone Class	Dose N Applied	NDVI_OTG
PFM	0.291	0.515	0.456
PDM	0.581	0.723	0.456
PNC	0.648	0.591	0.456
CP	0.648	0.591	0.456
CPyield	0.681	0.993	0.175
NUp	0.681	0.993	0.175
NDVI_OTG	0.562	0.116	-
UAV_I	0.888	0.799	0.278
UAV_II	0.678	0.380	0.456
NDVI_HMA	0.335	0.205	0.330

Significance codes: *** *p* < 0.001; ** *p* < 0.05; * *p* < 0.1. PFM—plant fresh matter (kh ha^–1^); PDM—plant dry matter (kg ha^–1^); PNC—plant nitrogen content (%); CP—crude protein (%); CPyield—crude protein per unit of area (kg ha^–1^); NUp—nitrogen uptake (kg ha^–1^); NDVI_OTG—NDVI of on-the-go sensor; UAV_I—NDVI of unmanned aircraft vehicle at first moment; UAV_II—NDVI of unmanned aircraft vehicle at second moment; NDVI_HMA—NDVI of handheld multispectral active sensor.

**Table 6 sensors-25-05811-t006:** Fertilizer doses applied per homogeneous zone and comparison with a fixed-rate application.

Classes	Trial	Area (ha)	Representativity (%)	Fertilizer Rate (kg ha^−1^)	Total Fertilizer Applied (kg ha^−1^)
-	Hypothetical fixed rate	1.5	100	150	225
1	Variable rate	0.002	0.14	90	0.180
2		0.017	1.16	105	1.785
3		0.741	50.34	120	88.920
4		0.739	50.20	135	99.765

**Table 7 sensors-25-05811-t007:** Characteristics of the optical sensors used in the study, including their classification, spatial resolution, and data footprint.

Sensor	Classification	Key Characteristic	Spatial Resolution	Digitization Footprint	Processing Effort/Time
HMA sensor	Active	Point sampling	50 cm^2^ per reading	98 Kb	1 h (interpolation and map generation)
UAV multispectral camera	Passive	Multi-band	5.3 cm/pixel	3 GB per ha	2 h (orthomosaic generation and NDVI calculation)
OTG sensor	Passive (sunlight sensing)	Real time, AI driven	12 pixel/cm (2.54 ppi)	106 Mb per ha	Instantaneous (processes on-the-go)

## Data Availability

The original contributions presented in the study are included in the article; further inquiries can be directed to the corresponding authors.

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
