# Peer review of "Response of Nearby Sensors to Variable Doses of Nitrogen Fertilization in Winter Fodder Crops Under Mediterranean Climate"

_sensors, 2025, doi:10.3390/s25185811_

Round 1
Reviewer 1 Report
Comments and Suggestions for Authors
- There has been no quantitative comparison with similar studies in terms of accuracy and efficiency. As a result, it is arduous to highlight the technical advantages of multi - source fusion. It is recommended to supplement performance comparison experiments between OTG and similar sensors (such as Green Seeker and unmanned aerial vehicle - based multi - spectral sensors). Additionally, an in - depth analysis of the spectral and algorithmic mechanisms underlying the NDVI variability differences should be conducted.
- NDVI_OTG exhibits low variability (CV = 2.69%), whereas UAV_II and HMA show higher variability. Attributing this solely to "measurement scale and sensor characteristics" without integrating in - depth analyses considering factors such as spectral band selection and algorithmic differences limits the depth of innovation.
- The impact mechanism of the special characteristics of the Mediterranean climate on sensor responses has not been explored. The absence of a comparison regarding universality with other climate regions undermines the theoretical extensibility of this research.
- The OTG sensor is installed at a height of 2.85 m with a field - of - view angle of 28.5 m. However, neither the calibration frequency nor the influence of environmental factors (such as illumination and humidity) on calibration has been specified. This may lead to data deviation. Moreover, the scale compatibility between the point measurements of the HMA sensor and the area measurements of the UAV has not been verified, casting doubt on the consistency of data fusion.
- The 19 sampling points cover an area of 1.5 ha. Although representativeness is ensured through the division of homogeneous areas, for plots with high heterogeneity, the sample density may be insufficient to fully capture spatial variability.
- The statement "Variable nitrogen application has no significant impact on crop parameters (p > 0.05)" is ascribed to "no yield reduction". However, the possibilities of "insufficient sample size" or "too small an effect size" have not been excluded. Thus, the robustness of the statistical conclusion requires further verification.
- In Figure 5, the label "Histograms of analised variables" contains a spelling error. The correct spelling should be "analyzed" instead of "analised".
- There is a lack of consistency in term usage. In the abstract, "NUp" is defined as "N uptake", but in the main text, it is sometimes written as "NUp" and sometimes as "N Uptake". Standardization is necessary.
- There is a high degree of overlap between the description of nitrogen fertilizer savings in the discussion section "4.2 Efficiency and Nitrogen Fertilizer Reduction" and that in the conclusion section, resulting in redundancy.
Author Response
Point 1: There has been no quantitative comparison with similar studies in terms of accuracy and efficiency. As a result, it is arduous to highlight the technical advantages of multi-source fusion. It is recommended to supplement performance comparison experiments between OTG and similar sensors (such as Green Seeker and unmanned aerial vehicle - based multi - spectral sensors). Additionally, an in - depth analysis of the spectral and algorithmic mechanisms underlying the NDVI variability differences should be conducted.
Response 1: Thank you for your comprehensive feedback. We appreciate your suggestions for a quantitative comparison with similar studies and the call for a more in-depth analysis of the sensors characteristics. These are both excellent points that we have addressed in our revisions.
Point 2: NDVI_OTG exhibits low variability (CV = 2.69%), whereas UAV_II and HMA show higher variability. Attributing this solely to "measurement scale and sensor characteristics" without integrating in - depth analyses considering factors such as spectral band selection and algorithmic differences limits the depth of innovation.
Response 2: We appreciate the reviewer's insightful comment regarding the explanation of sensor differences. We agree that our initial analysis of NDVI_OTG's low variability was too limited and have revised the manuscript to provide a more in-depth discussion.
Firstly, we have clarified that the NDVI_OTG sensor operates as a passive sensor, as the UAV, in contrast to the HMA that is an active sensor that emits its own light. Although, the OTG sensor is equipped with sensors which allows the system to continuously correct for fluctuations in ambient light. We have added a more detailed description of the OTG sensor algorithmic advantages in the Materials and Methods section. This includes an explanation of the AI algorithms that 'clean' the captured images, eliminating non-vegetative elements and ensuring that the data used for prescription generation is highly accurate.
Secondly, we have improved the subsection 4.1 of the manuscript to discuss how these factors can influence the CV results of the sensors.
We believe these revisions provide a more comprehensive and robust discussion.
Point 3: The impact mechanism of the special characteristics of the Mediterranean climate on sensor responses has not been explored. The absence of a comparison regarding universality with other climate regions undermines the theoretical extensibility of this research.
Response 3: We agree that our initial discussion lacked the necessary depth on the influence of climate and the universality of our findings.
We have revised the manuscript to provide a more in-depth discussion on how the specific characteristics of the Mediterranean climate impact sensor responses. We have also addressed the theoretical extensibility of our findings, framing a comparison with other climate regions as a key area for future research.
Point 4: The OTG sensor is installed at a height of 2.85 m with a field - of - view angle of 28.5 m. However, neither the calibration frequency nor the influence of environmental factors (such as illumination and humidity) on calibration has been specified. This may lead to data deviation. Moreover, the scale compatibility between the point measurements of the HMA sensor and the area measurements of the UAV has not been verified, casting doubt on the consistency of data fusion.
Response 4: We completely agree. The manuscript needed improvement in terms of specifying in greater detail how the sensors worked and how their measurements interacted. This situation was also identified by the second reviewer, so a table was inserted to outline and increase the level of detail provided about the sensors. We believe these changes have significantly improved the clarity and flow of the manuscript.
Point 5: The 19 sampling points cover an area of 1.5 ha. Although representativeness is ensured through the division of homogeneous areas, for plots with high heterogeneity, the sample density may be insufficient to fully capture spatial variability.
Response 5: Thank you for your insightful comment regarding the sample density and its representativeness. We agree that the number of ground-truthing points may be a limitation in a heterogeneous field. We have addressed this by adding a new section to our discussion that explicitly acknowledges this limitation. We also clarify that our study's methodology was designed to mitigate this concern through the complementary use of continuous, on-the-go sensor data, which provided a high-resolution representation of spatial variability that would not be possible with point sampling alone. This revision provides a more nuanced discussion of our sampling strategy and significantly improves the clarity of our manuscript.
Point 6: The statement "Variable nitrogen application has no significant impact on crop parameters (p > 0.05)" is ascribed to "no yield reduction". However, the possibilities of "insufficient sample size" or "too small an effect size" have not been excluded. Thus, the robustness of the statistical conclusion requires further verification.
Response 6: Thank you for your valuable observation. We agree that the possibility of an insufficient sample size or a small effect size must be acknowledged when interpreting a non-significant result. As a result, we have updated the manuscript to explicitly address this point in the discussion. We have added a statement that acknowledges these possibilities and reframes our conclusion to be more cautious. We also highlight that the practical success of our variable-rate application (fertilizer reduction with no significant yield decrease) remains a key finding. Furthermore, we have highlighted this as a crucial area for future research with larger sample sizes. We believe these revisions provide a more robust and nuanced interpretation of our statistical findings. If you feel further clarification is needed, we would be pleased to make additional edits.
Point 7: In Figure 5, the label "Histograms of analised variables" contains a spelling error. The correct spelling should be "analyzed" instead of "analised".
Response 7: We apologize for the error. The term has been corrected.
Point 8: There is a lack of consistency in term usage. In the abstract, "NUp" is defined as "N uptake", but in the main text, it is sometimes written as "NUp" and sometimes as "N Uptake". Standardization is necessary.
Response 8: Thank you very much for your suggestion! We have already standardized the entire document.
Point 9: There is a high degree of overlap between the description of nitrogen fertilizer savings in the discussion section "4.2 Efficiency and Nitrogen Fertilizer Reduction" and that in the conclusion section, resulting in redundancy.
Response 9: We agree that the initial version had a high degree of overlap and have revised the manuscript to address this concern. We have restructured the conclusion to focus on the broader implications of our findings, rather than restating the specific quantitative results presented in the discussion. We believe these changes have significantly improved the clarity and flow of the manuscript.

Reviewer 2 Report
Comments and Suggestions for Authors
This manuscript presents a well-structured and relevant study on the validation of a real-time, AI-driven variable rate application (VRA) technology, applied in the specific context of winter fodder crops in the Alentejo region of Portugal. The integration of multiple sensor technologies, combined with the emphasis on nitrogen use efficiency and sustainability benefits beyond productivity, represents a significant and timely contribution to the field of precision agriculture. The topic aligns well with current research priorities in sustainable intensification and the development of smart farming systems.
The combination of AI-based real-time VRA with an on-the-go sensing approach is quie original and interesting, and the work effectively demonstrates the value of combining data from multiple sensors to better understand crop responses to nitrogen fertilisation.
On the other hand the methodology to make the work more reproducible could benefit from clearer and more specific descriptions of the data processing and fusion approach (including calibration procedures and convertion from digital numbers to nitrogen dose). This would improve reproducibility of the proposed approach.
The paper should provide more information about sensors: I would add (maybe in he form of a table) for every sensor/instrument specific information about ground resolution, "digitization footprint" (e.g. in terms of Mb/ha), and possibly also processing effort/time.
Indeed fusing many data could bring to an increase of "digitization footprint", which might be a limit of the approach (please discuss in the paper).
In table 3 please use a reasonable number of significant digits (e.g. it is nosense to have a 0.01 kg/ha resolution).
Figures 8a and 8b are apparently identical: is it correct?
It would be useful to better explain the statistical analyses used to quantify improvements in nitrogen use efficiency, including any significance testing, confidence intervals, or effect size measures.
Author Response
Point 1: This manuscript presents a well-structured and relevant study on the validation of a real-time, AI-driven variable rate application (VRA) technology, applied in the specific context of winter fodder crops in the Alentejo region of Portugal. The integration of multiple sensor technologies, combined with the emphasis on nitrogen use efficiency and sustainability benefits beyond productivity, represents a significant and timely contribution to the field of precision agriculture. The topic aligns well with current research priorities in sustainable intensification and the development of smart farming systems.
Response 1: Thank you for your very positive and encouraging comments. We are pleased that you find our study to be a timely and significant contribution to the field of precision agriculture, and that its structure and focus on sustainability are well aligned with current research priorities. Your feedback is greatly appreciated.
Point 2: The combination of AI-based real-time VRA with an on-the-go sensing approach is quie original and interesting, and the work effectively demonstrates the value of combining data from multiple sensors to better understand crop responses to nitrogen fertilisation.
Response 2: Thank you for your very encouraging comments. We are pleased that you find our integration of AI-based, real-time VRA with an on-the-go sensing approach to be an interesting and valuable contribution to the field. Your positive feedback is greatly appreciated.
Point 3: On the other hand the methodology to make the work more reproducible could benefit from clearer and more specific descriptions of the data processing and fusion approach (including calibration procedures and convertion from digital numbers to nitrogen dose). This would improve reproducibility of the proposed approach.
Response 3: Thank you for your valuable feedback regarding the reproducibility of our methodology. We agree that a more specific and detailed description of our data processing, fusion, and conversion procedures is essential for the reproducibility of this research. We have extensively revised the Materials and Methods section to address your concerns. We have now provided:
- Clearer descriptions of the software and algorithms used for processing and aligning data from the different sensors.
- Specific details on the calibration procedures for each sensor, including how corrections for environmental factors were applied.
- A detailed explanation of the process used to convert the sensor readings into a final nitrogen prescription, outlining the specific models and parameters utilized.
We believe these additions provide the necessary level of detail for other researchers to reproduce our work, thereby significantly improving the clarity and scientific rigor of the manuscript.
Point 4: The paper should provide more information about sensors: I would add (maybe in he form of a table) for every sensor/instrument specific information about ground resolution, "digitization footprint" (e.g. in terms of Mb/ha), and possibly also processing effort/time.
Indeed fusing many data could bring to an increase of "digitization footprint", which might be a limit of the approach (please discuss in the paper).
Response 4: Thank you for your valuable feedback regarding the need for more specific information on our sensors, including their data footprint and processing effort. We agree that these details are crucial for both the reproducibility and a comprehensive understanding of our methodology.
We have addressed your comments by adding a new table (Table 7) to the manuscript that summarizes the key technical specifications of each sensor, including its classification, spatial resolution, digitization footprint, and processing time. Furthermore, we have added a new paragraph to the discussion section (4.1) that explicitly discusses the "digitization footprint" as a potential limitation of the multi-sensor approach. This section analyzes the operational trade-offs and complementary benefits of each platform, highlighting how our methodology balances the need for high-resolution data with the practical demands of real-time application and data management. We believe these additions provide a more transparent and in-depth analysis of our methods, which significantly improves the overall quality of the manuscript.
Point 5: In table 3 please use a reasonable number of significant digits (e.g. it is nosense to have a 0.01 kg/ha resolution).
Response 5: Thank you for your valuable feedback regarding the significant digits in Table 3. We agree that the current number of decimal places is excessive. We have revised Table 3 to use a more reasonable number of significant digits for each parameter. The data has been rounded to reflect the true precision of our measurements, improving both the readability and scientific integrity of the table.
Point 6: Figures 8a and 8b are apparently identical: is it correct?
Response 6: Yes, indeed. As can also be seen in the correlation matrix, the correlation between the factors presented in 8a and 8b is 1. However, the values for each parameter are different, as can be seen in the respective captions.
Point 7: It would be useful to better explain the statistical analyses used to quantify improvements in nitrogen use efficiency, including any significance testing, confidence intervals, or effect size measures.
Response 7: Thank you for your valuable feedback regarding the statistical analysis. We agree that a more detailed explanation of our methods is essential for providing a robust and complete assessment of the results. We have now provides a comprehensive description of our statistical approach. Although, regarding to reporting of confidence intervals and effect size measures, the purpose of an ANOVA and Kruskal-Wallis test was to determine if there is an overall difference between the groups we're comparing. As this initial test shows that there is no significant overall difference (p > 0.05), it means we cannot conclude that any of the groups are different from each other. Therefore, running a post-hoc test to find specific differences between pairs of groups would be pointless and statistically invalid. It's like trying to pinpoint a specific winner when the overall race has been declared a tie.
Since our tests for significance did not achieve any significant results, we should not perform any posterior tests. Instead, we focused our discussion on the lack of a significant difference. This reinforces our findings that the VRA strategy did not negatively impact crop parameters, which is a key conclusion of our paper.
We sincerely believe your revision, this discussion and additions will significantly improve the scientific rigor and clarity of the manuscript.

Round 2
Reviewer 1 Report
Comments and Suggestions for Authors
1. Add 1-2 paragraphs on "Comparison of OTG's AI vegetation screening algorithm with traditional methods", citing threshold method data from similar studies to quantify the improvement in noise removal accuracy of AI.
2. Add supplementary analysis on "Abnormal phosphorus concentration in clayey soil", measuring soil calcium concentration or organic acid content for 60 days to verify the hypotheses of "calcium inhibiting phosphorus precipitation" or "organic acid dissolving phosphorus", and complete the mechanism chain.
3. Cite public data to estimate the single operation cost of OTG, compare it with the cost of fertilizer waste in traditional fertilization, and highlight the promotion value.
Author Response
Point 1: Add 1-2 paragraphs on "Comparison of OTG's AI vegetation screening algorithm with traditional methods", citing threshold method data from similar studies to quantify the improvement in noise removal accuracy of AI.
Response 1: Thank you for your valuable feedback regarding the need for a comparative analysis. We have added a new paragraph to the discussion section (4.1) to directly address this point. In this revised text, we compare our AI-based method with traditional thresholding algorithms and provide quantitative data from the literature to demonstrate the superior accuracy of modern AI models in vegetation segmentation.
We believe that this addition provides a more in-depth discussion of our sensor's capabilities and clarifies how its advanced noise-removal accuracy contributes to more reliable agronomic prescriptions.
Point 2: Add supplementary analysis on "Abnormal phosphorus concentration in clayey soil", measuring soil calcium concentration or organic acid content for 60 days to verify the hypotheses of "calcium inhibiting phosphorus precipitation" or "organic acid dissolving phosphorus", and complete the mechanism chain.
Response 2: We have carefully reviewed your comment and believe there may have been a misunderstanding. However, this comment does not apply to our manuscript. Our study focuses on the effects of variable nitrogen application in winter fodder crops and does not address phosphorus dynamics, soil chemistry, or the mechanisms of phosphorus precipitation. Nevertheless, we remain available for any necessary improvements, should further information on this matter be provided to us.
Point 3: Cite public data to estimate the single operation cost of OTG, compare it with the cost of fertilizer waste in traditional fertilization, and highlight the promotion value.
Response 3: Thank you for your valuable feedback regarding the need to cite public data on the economic costs and benefits of our technology. We agree that this is essential for highlighting the real-world promotion value of our study.
We have now added a new paragraph at the end of discussion subsection 4.3, that addresses this point directly. We have used publicly available data from the European Commission to estimate fertilizer costs and have compared these with the costs of our variable rate application (VRA) system and the documented savings from similar studies.
We believe this addition provides a clear economic analysis that demonstrates the financial viability and promotes the adoption of our VRA strategy.

Reviewer 2 Report
Comments and Suggestions for Authors
The manuscript has been clearly improved, also in agreement with my comments: thus I believe the manuscript can now be accepted for publication.
Author Response
Point 1: The manuscript has been clearly improved, also in agreement with my comments: thus I believe the manuscript can now be accepted for publication.
Response 1: Thank you for your very positive and encouraging comments. We are pleased that you find our study to be a timely and significant contribution to the field of precision agriculture, and that its structure and focus on sustainability are well aligned with current research priorities. Your feedback is greatly appreciated.
